# The Impact of the Nervous System on Arteries and the Heart: The Neuroimmune Cardiovascular Circuit Hypothesis

**DOI:** 10.3390/cells12202485

**Published:** 2023-10-19

**Authors:** Sarajo K. Mohanta, Ting Sun, Shu Lu, Zhihua Wang, Xi Zhang, Changjun Yin, Christian Weber, Andreas J. R. Habenicht

**Affiliations:** 1Institute for Cardiovascular Prevention, Ludwig-Maximilians-Universität (LMU) München, 80336 Munich, Germany; ting.sun@med.uni-muenchen.de (T.S.); shu.lu@med.uni-muenchen.de (S.L.); zhihua.wang@med.uni-muenchen.de (Z.W.); xi.zhang@med.uni-muenchen.de (X.Z.); changjun.yin@med.uni-muenchen.de (C.Y.); christian.weber@med.uni-muenchen.de (C.W.); 2German Center for Cardiovascular Research (DZHK), Partner Site Munich Heart Alliance, 80336 Munich, Germany; 3Easemedcontrol R&D, Schraudolphstraße 5, 80799 Munich, Germany; 4Institute of Precision Medicine, The First Affiliated Hospital of Sun Yat-sen University, Guangzhou 510030, China

**Keywords:** atherosclerosis, artery, heart, brain, neuroimmune interactions

## Abstract

Three systemic biological systems, i.e., the nervous, the immune, and the cardiovascular systems, form a mutually responsive and forward-acting tissue network to regulate acute and chronic cardiovascular function in health and disease. Two sub-circuits within the cardiovascular system have been described, the artery brain circuit (ABC) and the heart brain circuit (HBC), forming a large cardiovascular brain circuit (CBC). Likewise, the nervous system consists of the peripheral nervous system and the central nervous system with their functional distinct sensory and effector arms. Moreover, the immune system with its constituents, i.e., the innate and the adaptive immune systems, interact with the CBC and the nervous system at multiple levels. As understanding the structure and inner workings of the CBC gains momentum, it becomes evident that further research into the CBC may lead to unprecedented classes of therapies to treat cardiovascular diseases as multiple new biologically active molecules are being discovered that likely affect cardiovascular disease progression. Here, we weigh the merits of integrating these recent observations in cardiovascular neurobiology into previous views of cardiovascular disease pathogeneses. These considerations lead us to propose the Neuroimmune Cardiovascular Circuit Hypothesis.

## 1. Introduction

Recent studies have made considerable progress in understanding the tripartite interaction networks of the immune system, the nervous system, and the cardiovascular system [1]. Neuroimmune cardiovascular crosstalk-related research aims to anatomically and functionally define the multitude of biologically and clinically important networks of cell-derived signals and their cognate receptors originating in neurons, immune cells, and cells of the cardiovascular system, including endothelial cells, smooth muscle cells, adventitial cells, and multiple heart cells [2], to maintain local and systemic tissue homeostasis through multiple neural circuits of the body–brain and brain–body axes. Moreover, each of the cells in this broad pan-tissue network is equipped with neuronal sensory receptors outside the CBC such as the skin, the gastrointestinal tract, the lung, and the endocrine system, and all are targets of the effector arms of the circuits [1]. Neural circuits, broadly speaking, consist of afferent sensory arms that innervate peripheral tissues and efferent effector arms that project back to the peripheral tissue after being modified in brain integration centers [3,4,5,6,7]. Progress made in this area may lead to new experimental approaches to examine the mediators for the maintenance of physiologically important aspects of cardiovascular homeostasis. However, bona fide neural brain circuits are not organized as simple neural reflexes, instead various forms of all-important brain integration centers are critically involved in receiving information from both the peripheral organs and higher brain areas [8,9]. Accordingly, integration centers not only receive signals from body-derived sensory axons and the peripheral endocrine system in feedback loops, but also from inputs originating in other brain areas to eventually shape the efferent brain–body axes, including memories stored in distinct brain nuclei and indeed integration centers [8,10,11]. Thus, neural brain circuits are large neuronal networks composed of highly dynamic and flexible afferents, integration centers, and efferent output projections. The overall functional implications for disease pathogeneses of neural brain circuits are countless, ranging to respond to immediate survival needs such as danger avoidance, heat, blood pressure, response to hunger and thirst, but also to processing long-term brain traits such as many types of memory, chronic pain adjustments, organ injury, and even strategic thinking and planning [12,13]. While this article cannot comprehensively cover the enormous recent progress to understand integration centers, several basic principles require attention.

The brain can be viewed as a storage hub of trillions of neurons tasked to control peripheral tissue homeostasis [14,15,16]. Brain neurons are anatomically organized in temporal and functional hierarchical patterns in engrams to regulate homeodynamics and adaptability, especially in association with learning, memory, and the transfer of information [17,18]. Located between incoming interoceptive sensory information and outgoing efferent signals, the impact of integration centers can be, in short, described as complex networks of brain neurons involved in supporting the survival of multicellular organisms during multiple acute or chronic forms of environmental and internal body challenges (also referred to as stress-signals by some investigators) [11,19,20]. Importantly, our understanding of memory has changed and now includes immune memory and, more broadly, tissue disease memory: the latter type of memory have been shown to be engraved into distinct central nervous system neurons that memorize a previous organ injury in which the immune system and soluble mediators had been involved [6,21]. Moreover, tissue disease memory may also include epigenetic changes in neurons and in non-neuronal tissues [22]. All these inputs are eventually received by neurons in the integration centers and processed together with the sensory information to modify the effector arm of the circuits. A more extensive review of the CBC has recently been published by us [1]. Thus, the complexity of the integration center-type neuronal networks in the brain can be defined as both function- and anatomy-defined engrams that involve neuronal subtypes located in different brain nuclei whose structural anatomy are continuously changing in very short- (seconds) or long-term (years; decades) time windows [1]. Therefore, the resulting brain activity to support the survival of the host can only be met by flexible and rapidly acting adaptation mechanisms that respond to inputs from the internal organs, the endocrine system, and brain engrams [21,23,24,25,26,27] to form the efferent neuronal signals in each individual.

Neural brain circuits have recently been characterized involving tissues as varied as adipose tissue [28,29], the immune system [3,4], the heart [30,31,32], the lung [33], and the gut [4,26], and diseases [3,28,29] as wide-ranging as cancers [13,34], infectious diseases [33], and autoimmune diseases [8,13]. Moreover, two recently identified brain circuits within the cardiovascular system, i.e., the ABC [35] and the HBC [36,37], interact to form a large CBC [1]. Here, we will review the evidence that a CBC exists and summarize the circumstantial evidence that the CBC is critically involved in the control of cardiovascular diseases (CVDs). Delineating the CBC requires molecular, structural, and functional information of the interactions and connections of the peripheral nervous system and the central nervous system within the ABC and the HBC, in addition to their interactions with numerous peripheral extra-cardiovascular tissues [1,3,4,5,8,15,38]. By highlighting critical features of the CBC, we propose a novel wide-ranging and holistic hypothesis of CVD pathogenesis that we have termed the Neuroimmune Cardiovascular Circuit Hypothesis [1].

## 2. Tripartite Interaction between the Nervous System, the Immune System, and the Cardiovascular System Gives Rise to the Neuroimmune Cardiovascular Interface Concept

The father of modern anatomy Andreas Vesalius in his book *De Humani Corporis Fabrica* in the sixteenth century reported the close physical interaction between the nervous and cardiovascular systems (reviewed in [39]). In the late nineteenth and early twentieth centuries, Goltz and Stricker studied the relationships between the nervous and the immune systems [40]. The analyses of these systemically interacting biological systems developed into independent research areas, i.e., the area of the neurovascular link and the ever expanding area of neuroimmunology, respectively [1,3,8]. Neuroimmune crosstalks are known to maintain tissue homeostasis. Peripheral inflammation is sensed and affected by the nervous system [3,4,8]. However, the relation between atherosclerosis and the nervous system remained—until recently—largely unknown. We reported that the peripheral nervous system uses the adventitia of all arteries as its main conduit to reach all tissues, as originally observed by Vesalius at a macroanatomical scale [35,39].

On the basis of these studies, we proposed that three systems rather than two, i.e., the cardiovascular system, the immune system, and the nervous system, reciprocally interact in a tripartite manner and form neuroimmune cardiovascular interfaces [35] (Figure 1). It is noteworthy that these bidirectional interaction patterns (the intensity of each information transfer or the way of the interactions) can be contextualized to different conditions including genetically conditioned healthy states, and behavioral or disease states, including altered ligand–receptor pair interactions [41,42]. Thus, the bidirectional interaction patterns in neuro-cardiovascular settings in humans are different between each healthy individual (since they are genetically diverse), between different diseases (since their risk factor profiles are variable), and between different behavioral conditions (since their education, and socializations are distinct). These studies—when integrated into traditional views of current cardiovascular disease concepts [35,36]—lead us to propose the Neuroimmune Cardiovascular Circuit Hypothesis (Figure 2).

## 3. Neural Circuits of the Cardiovascular System

### 3.1. The ABC

The ABC consists of all components of the peripheral nervous system that differentially innervate different segments of the arterial tree, and other components of the vascular system such as lymph vessels, high endothelial venules in secondary lymphoid tissues, veins, and others, in territory-specific ways [30,35,43,44,45]. The ABC emerges as heterogeneous networks of peripheral nervous system connections with its major component, i.e., the dorsal root ganglia, and nodose ganglia as sensors, and the cranial nerves and sympathetic ganglia as effectors [12,30,35]. New features of the peripheral nervous system innervation of the ABC including the direct innervation of medium- and large-sized arteries were discovered following the anatomical and cellular characterization of neuroimmune cardiovascular interfaces in the aorta adventitia in atherosclerotic mice and human cardiovascular tissues [35,46,47]. It is noteworthy, however, that the peripheral nervous system forms multiple innervation patterns in different parts of the cardiovascular system, with different constituents depending on local needs to sense normal physiological and pathophysiological changes [30,35]. Major aspects of neuroimmune cardiovascular interfaces and additional innervation patterns that describe the principle features of the ABC and the HBC have recently been reviewed by us, including neurotransmitters, chemokines, and sensory receptors [35].

### 3.2. The HBC

The HBC is the better known companion of the CBC [32,48]. Unlike the ABC, the HBC has been characterized in considerable detail, involving several peripheral nervous system components: a major sensory vagal neuron component, several efferent vagal and sympathetic nervous system components, together with a relatively independently acting intrinsic nervous system in the heart [31,43,49,50,51,52]. As the HBC involves sensors that receive signals as varied as sensing atrial blood filling, aorta blood pressure, heart failure of the left ventricle, and breathing rhythms of the lung, it is apparent that risk factors of CVDs including hypertension [38], high lipid levels [53,54], coronary artery disease [53,55], healing following a myocardial infarct [56,57], diabetes mellitus [58,59,60], and obesity [28,29,61,62] are candidates for crosstalk between the ABC and the HBC, though the precise mechanisms remain to be identified and are likely complex. The challenges to define these crosstalks at cellular, molecular, and functional levels are enormous, requiring all tools of basic research as well as considerable efforts to conduct translational studies. The HBC has recently been reviewed and, therefore, here, will not be considered further [35].

### 3.3. The CBC

Defining and delineating the CBC with its major circuit components, i.e., the ABC and the HBC, became possible through the recent identification of a neuroimmune cardiovascular interface-triggered structural ABC [35,63]. It is well established that there are multiple bidirectional interactions between the arteries and the heart in physiology and pathophysiology, forming a multilayered and large CBC network [12,43,64]. These interactions are particularly apparent when risk factors of atherosclerosis are considered: hypertension is sensed by various neurons of the HBC including those sensing atrial filling, aorta distension, and left ventricular failure leading to tachycardia, possibly supraventricular arrhythmias, and these neuronal activities may lead to a reduced blood supply and the perfusion of vital organs including the heart, the kidney, and the brain [12,30,35,36,38,43]. In advanced stages of coronary heart disease, the oxygen supply to the myocardium is severely impaired, aggravating left ventricular failure, and the list of direct interactions between the ABC and the HBC to form the CBC can be continued. Major unanswered questions relate to the identity of brain neurons that receive sensory inputs from arteries and the heart, respectively: whether the integration of both sub-circuits alter the neuronal activity in brain integration centers (Figure 2) and how the efferent arms of the ABC versus the HBC are altered in the periphery by the integration of signals projected to the brain. Moreover, the advanced atherosclerosis of the brain vasculature may affect the supply of nutrients and oxygen to the brain, and thereby directly affects the brain via multiple mechanisms before clinically significant strokes may develop. Therefore, the less well understood phenomenon of vascular dementia [65] may be affected by numerous inputs from the ABC and the HBC. The delineation of the inner workings of the CBC may lead to a better understanding of vascular dementia and its relation to degenerative brain diseases including Alzheimer’s disease [1,66,67]. There are several limitations to our understanding of the CBC: Most studies have been performed in mice and the translation of the most significant findings in mice to humans may take decades. Moreover, intervention into the nervous system in humans may not be achieved soon as the preclinical array of evidence must be worked out in considerably more detail first, followed by small clinical trials of high-risk patients before large prospective clinical trials can be initiated.

## 4. Interoception and Exteroception Involve Participation of Risk Factors for CVDs

Neurobiology distinguishes signals that are derived from the body (interoception), including the genetic makeup of each individual and those that are derived from the environment (exteroception) including heat, cold, bacterial infections, and numerous environmentally defined cues (Figure 2). While the genetic makeup in a given study of experimental mice is identical, each human owns its individual interoceptive pathways based on the combination of functionally relevant genetic mutations and the acquired interoceptive brain engrams during their lifetimes [21,25,26,36]. The interoceptive pathways can either prevent or accelerate diseases, as shown for a series of genetic mutations that affect diseases as diverse as multiple types of cancer, disease sensitivities due to mutations of genes controlling the immune system, but also CVDs and dementia, such as mutations in the transport of lipids (Apolipoprotein E isoforms; low-density lipoprotein receptor mutations and other lipid transport-relevant genes are examples of numerous mutations affecting the cardiovascular system), and more will be discovered through genome-wide association studies [68,69]. On the other side of the interoception–exteroception pair, body surfaces including the skin, the lung, and the gut harbor sensory receptors such as members of the transient receptor potential channel V1 that had been discovered as a temperature receptor by David Julius [70,71], and other members of the transient receptor potential family, in addition to Piezo1 and Piezo2 serving as stretch receptors, discovered by Ardem Patapoutian [43]. Many of these inputs project to the brain stem [33]. Exteroception pathways that impact the CBC and, in particular, CVD pathogenesis clearly need to consider risk factor profiles, including those rendered by dietary habits, body weight, sedentary lifestyles, smoking, and multiple types of disease-promoting forms of environmental and internal body challenges that are referred to stress by some scholars [38,57,72,73,74].

## 5. Why Propose the Neuroimmune Cardiovascular Circuit Hypothesis Now?

Various hypotheses including the response to injury hypothesis, the lipid storage hypothesis, the immune/inflammation hypothesis, among others have been proposed to guide research into the pathogenesis of CVDs [53,54,55,57,75]. However, many of these hypotheses did not lead to established clinical treatments except for lipid-lowering drugs [1]. However, some inflammation-inhibitory drugs including colchicine are being evaluated in clinical trials, but whether they reach a more advanced treatment stage remains to be seen [76]. Recent observations suggest that the regulation of CVDs involves and is strongly affected by the nervous system with its instructor and control master, i.e., the brain [6,21,26,35,36,38,43,51,77]. The nervous system interacts with the immune system in bipartite and mutual ways to maintain physiological homeostasis in the arterial tree and the heart [1,78,79]. In addition, ABC and HBC circuitries interact to form a CBC network [1,35,36]. Comprehension of the CBC is still incomplete and much needs to be learned from the interaction of the ABC and the HBC. Here, we propose a new inclusive and holistic hypothesis of the pathogenesis of the cardiovascular system that we term as the Neuroimmune Cardiovascular Circuit Hypothesis: the brain is viewed as the primary center in which both genes and intero-/exteroceptive cues are sensed by brain integration centers that are organized in engrams [23]. Additionally, the hypothesis incorporates previous hypotheses as these describe exteroceptive and interoceptive cues that target the cardiovascular system. We anticipate that this hypothesis may help to comprehend and make progress in understanding the complex nature of cardiovascular diseases as regulated by the nervous system. Neuroimmune interactions have recently been reviewed by us [1]. It should be noted that the neuroimmune cardiovascular circuit are only beginning to emerge. Moreover, although the amygdala and other brain nuclei that we identified in atherosclerotic mice vs WT brains are known to serve as integration centers via the pain pathway [9,35], much needs to be learned from various mouse models regarding the precise neuron subtypes in order to define the integration center networks and their input and output signals [1,12,13,23]. It is apparent that most of the clinical potential of the mouse data are largely hypothetical because translational work is only beginning. Future directions should be aimed at using human brain imaging in patients with various advanced stages of cardiovascular diseases including ischemic heart disease post-myocardial infarcts using functional magnetic resonance imaging approaches in both acute cardiovascular events and chronic disease conditions.

## 6. Conclusions and Future Directions

The brain acts as a dynamic and powerful control center to regulate both physiological and pathophysiological responses within the CBC. Unlike experimental mice, human population genetics show an enormously heterogeneous genetic makeup that provides a large number of CVD-promoting or CVD-protective mutations. Some of these mutations are gender-sensitive. Genome-wide association studies have provided evidence of the scale of the heterogeneity of CVD-affecting mutations, but also mutations that affect brain traits which may involve degenerative brain diseases such as Alzheimer´s disease [66]. Moreover, the nervous system, which receives signals through interoceptive and exteroceptive projections, organizes itself in the form of only partially understood brain engrams [1]. There is no well understood CVD-specific engram available yet, but related engrams have been proposed for immune memory, thirst and stress-related engrams, among others. To identify CVD engrams in various mouse models, future studies will lead the way to initiate tangible translational clinical studies in humans. Combining experimental studies with translational research may therefore lead to a better understanding of the role of the brain and its neuron subsets to control CVD outcomes in humans. The brain as a control hub to direct organ function in the periphery, including arteries and the heart, adds a major complexity to the analyses of CVDs in both experimental mice and clinical trials. Translational studies in humans require functional assays to image human brain function during disease progression, including functional magnetic resonance imaging approaches. Further progress in genome-wide association study-related works, and molecular and structural studies of human brains are needed [80,81]. Studies of the cardiovascular and immune systems together with progress in the neurobiology of the CBC may lead to the identification of new clinically important checkpoints in the nervous and cardiovascular systems that affect disease progression. When the genetic makeup is considered with the exteroception-triggered inputs, a comprehensive hypothesis regarding CVD progression may be considered. We suggest to designate this hypothesis the *Neuroimmune Cardiovascular Circuit Hypothesis* (Figure 2).

## Figures and Tables

**Figure 1 cells-12-02485-f001:**
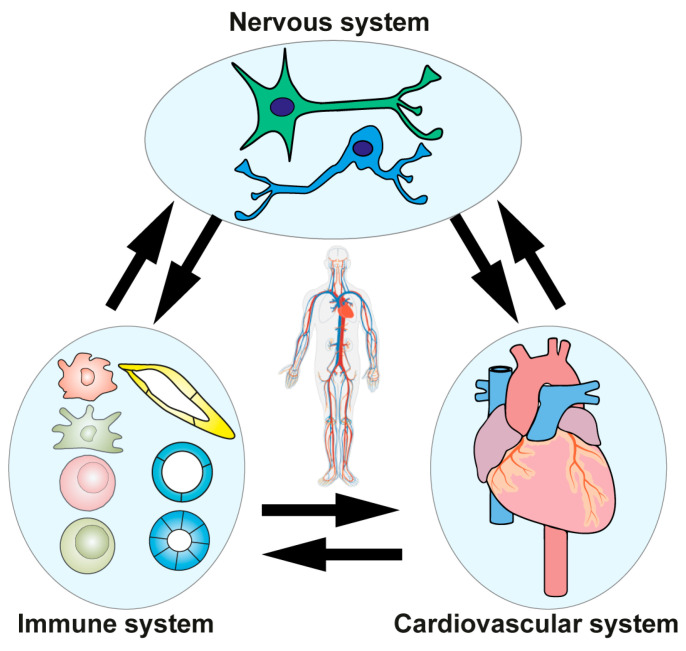
A tale of three systems: Neuroimmune cardiovascular interfaces form in normal and diseased cardiovascular tissues. The nervous, immune, and cardiovascular systems form an interaction triad where mutually interacting tissues communicate with each other in bidirectional as well as in tridirectional ways to regulate cardiovascular function during normal physiology and, more so, during diseases in humans. The nervous system interacts with the immune system to establish hard-wired sensory and effector arms of peripheral tissue-derived inputs and regulate the brain output signals derived from integration centers. The immune and cardiovascular systems interact to regulate inflammation and maintain tissue homeostasis. Likewise, the nervous and cardiovascular systems interact at multiple levels and share developmental and homeostatic cues in physiology. We have named these anatomically and functionally definable tripartite interactions as neuroimmune cardiovascular interfaces.

**Figure 2 cells-12-02485-f002:**
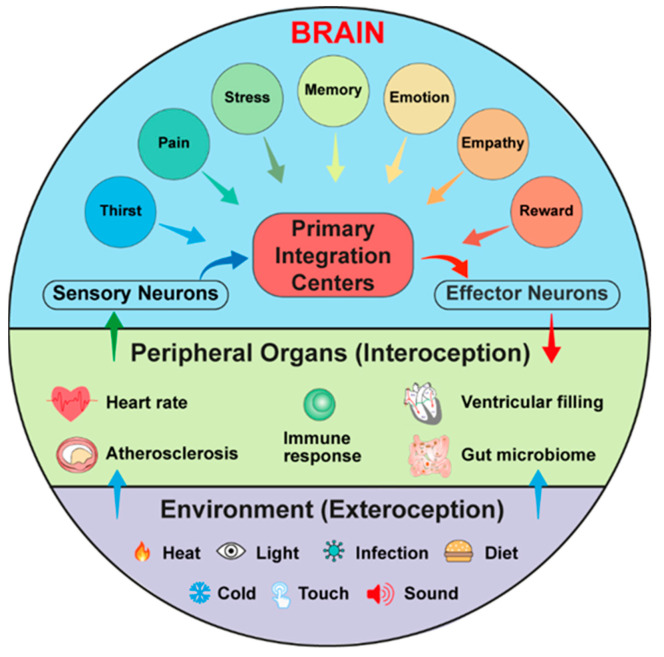
The brain controls the cardiovascular system via CBC neuronal exteroceptive and interoceptive networks consisting of multiple anatomically defined brain nuclei. The CBC can be described as a multilayered afferent and efferent axonal connection network with the brain at its central control center. The CBC sensory afferents receive the exteroceptive, i.e., environmentally-triggered signals such as heat, cold, touch, infection, light, and sound, either directly or via interoceptive signals that have been modified by the exteroceptive pathways. Thus, exteroception connects the environment with all body tissues including the endocrine system which, in turn, releases hormones to instruct the immune system to respond to numerous needs that impact traits such as metabolism, heart rate, and other tissue responses. Primary brain integration centers receive and then process sensory information from both peripheral tissues and -independently- from brain nuclei (located in the cortex or other higher brain nuclei) to shape the efferent arms of the CBC. Note that the primary integration centers that are connected to engrams are constantly changing and that the one shown here does not represent an anatomically defined constituent of the CBC, but rather designates a conceptual view. Thus, primary integration centers are large brain networks that receive a multitude of acute and chronically conveyed signals from numerous brain nuclei that are organized as engrams, including memory, chronic pain, stress, emotions, and empathy among many others.

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
