# Peer review of "The Impact of the Nervous System on Arteries and the Heart: The Neuroimmune Cardiovascular Circuit Hypothesis"

_cells, 2023, doi:10.3390/cells12202485_

Round 1
Reviewer 1 Report
The impact of the nervous and immune systems on arteries and the heart is commonly known an this is the topic of many review articles. What are new aspects in the manuscript presented by the authors?
The abstract is too short and enigmatic. I suggest to describe the subject of manuscript in more detail.
The importance of the topic should be emphasized in the introduction
The information about history of anatomy (page , lines 3-10) is interesting, but not directly related to the topic of the manuscript
Some abbreviations used in the article make it hard to read. For example PNS or CVDs. I suggest to use full phrase, not abbreviation for these examples.
In my opinion the roles of particular neurotransmitters/ neuromodulators in regulation of arteries and the heart functions is important aspect and should be added into the manuscript.
Clinically aspects of the observation described in the manuscript should be unserlined.
Conclusions are not clear. I suggest to reedit this part of the article
The authors cited relatively large number of their previous review articles and commentaries For example ref no 3, 4, 25). In my opinion they should be based first of all on original articles, not review articles
Author Response
Responses to REVIEWER 1:
R1: The impact of the nervous and immune systems on arteries and the heart is commonly known an this is the topic of many review articles. What are new aspects in the manuscript presented by the authors?
RESPONSE: Reviewer #1 rightfully asks that new aspects in the current review should be clearly spoken vs previous reviews by others. We made major attempts to meet this request in the revised manuscript. There are several new aspects that distinguishes our current review from previous reviews: i. The current review - on the basis of recent discoveries in the area of atherosclerosis since 2022 - emphasizes the need to analyze three biological systems, i.e. the immune system, the nervous system (this pair of biological systems focuses on immune cells as they interact with the nervous system) and the cardiovascular system (which largely consists of the arterial tree and the heart) in tripartite rather than - separately - in bipartite interactions. This concept became more visible and supportable with the discovery of the artery brain circuit (ABC) following our unexpected discovery of the direct involvement of artery-adventitia-nervous system where immune cells accumulate: a hardwired uninterrupted peripheral sensory nervous system arm whose dorsal root ganglia (DRG) axons enter the spinal cord from where it is projected to the brain; the identification of activated c-fos+ central nervous system neurons, the concept of integration centers and the description of a functional efferent arm consisting of the vagus nerve (Mohanta et al. Nature 2022); ii. our current review covers additional aspects of the ABC including the concept of engrams in the brain involving memory, the importance of integration centers and more (more extensively reviewed by Mohanta in 2023 (ref # 1); iii. the involvement of the ABC interactions gives rise to the neuroimmune cardiovascular circuit hypothesis involving both the arterial tree and the heart both of which interact with the nervous system and the immune system in complex networks. We put emphasis on the cardiovascular brain circuit (CBC) forming a biological system involving both the ABC and heart brain circuit (HBC) as outlined schematically in Figure 1 below; iv. our current review makes also an attempt to integrate the recent discoveries in the neurobiology of the cardiovascular system and multiple hypotheses involving lifestyles, oxidation reactions of lipoproteins, diet, genetic risk factors and other forms of exteroception (Figure 2 of the current review).
R1 continues: The abstract is too short and enigmatic. I suggest to describe the subject of manuscript in more detail.
RESPONSE: We have now extended the abstract as requested by Reviewer #1.
R1: The importance of the topic should be emphasized in the introduction.
RESPONSE: We now have emphasized the importance of the topic at the beginning of the introduction.
R1: Some abbreviations used in the article make it hard to read. For example PNS or CVDs. I suggest to use full phrase, not abbreviation for these examples.
RESPONSE: We replaced all “PNS abbreviations” as suggested by the reviewer. However, since CVD was abbreviated 16 times in the manuscript we chose to not replace CVD with full phrase.
R1: The information about history of anatomy (page , lines 3-10) is interesting, but not directly related to the topic of the manuscript.
RESPONSE: Since the review is based on the interaction of nervous, immune and cardiovascular system, we believe that a rather brief information on the founder of modern anatomy, i.e. Andreas Vesalius, whose discoveries led to our current neuroimmune cardiovascular circuit hypothesis may interest the readers. For these reasons, we would like to maintain these short remarks.
R1: Clinically aspects of the observation described in the manuscript should be unserlined.
RESPONSE: We now underlined the clinical aspects. However, the editors should make a decision whether this is consistent with their rules and opinion.
R1: In my opinion the roles of particular neurotransmitters/ neuromodulators in regulation of arteries and the heart functions is important aspect and should be added into the manuscript.
RESPONSE: Thank you for this important suggestion. Since the roles of specific neuromodulators on artery and heart functions were recently reviewed by us, which is now ref #1, we did not include them in this manuscript because of space constraints and to avoid duplication. However, we refer to this more comprehensive review in the current short review.
R1: Conclusions are not clear. I suggest to reedit this part of the article.
RESPONSE: we made attempts to more concisely summarize the conclusions.
R1: The authors cited relatively large number of their previous review articles and commentaries For example ref no 3, 4, 25). In my opinion they should be based first of all on original articles, not review articles.
RESPONSE: We removed ref # 3 and # 4, but kept the previous ref #25 (now ref #1) as it is closely related to the current manuscript.
Reviewer 2 Report
The review paper "The Impact of the Nervous System on Arteries and the Heart: The Neuroimmune Cardiovascular Circuit Hypothesis", written by authors Mohanta SK, Sun T, Wang Z, Zhang X, Yin Z, Weber C, Habenicht AJR is up-to-date, comprehensive and scientifically valuable contribution for integrative approach to understanding of cardiovascular physiology and diseases. Concept of "neurocardiovascular (diseases, circuits...)" is successfully substituted by "neuroimmune cardiovascular (circuit hypothesis)".
Major revisions
Principles like brain morpho-functional organization were discussed. "...the brain can be viewed as a storage hub of trillions of neurons tasked to control peripheral tissue homeostasis." (page 2/12, lines 4 &5) . It is my opinion that also the concept of "homeodynamics" and/or adaptability (Goldberger AD, 2006, doi 10.1513/pats200603-028MS; Matić Z et al, 2020, doi 10.3389/fphys.2020.00024) should be mentioned and discussed, especially in association with learning, memory and transfer of information.
In the next sentence (page 2/12, line 5-9), integration centers were described as neural networks supporting the survival of multicellular organisms "during multiple acute and chronic stress". It is my opinion that the term "stress" should be substituted by the term "environmental and internal body challenges". Modern definitions of stress include coping capacity of the organism, i.e. stress as the state of the organism exposed to environmental demands BEYOND coping capacity of the organism.
Concept of bidirectional (i.e. reciprocal) tripartite neuroimmune cardiovascular circuit is is very well presented as PRINCIPLE. Still, an important issue of PATTERN, individually specific for healthy individuals (genetically conditioned), behaviorally specific (Porta A et al, 2012, doi 10.1016/j.compbiomed.2011.04.019; Matić Z et al, 2020, doi 10.3389/fphys.2020.00024) or disease specific (Radovanović N et al., 2018, doi 10.3389/fphys.2018.00165) is missed to be mentioned. The pattern (i.e. intensity of the each information transfer within bidirectional communication) deserves to me mentioned.
Minor revisions and comments
Please make a new paragraph between figure legend and the following body text. After both figure legends text proceeds in continuation with the text of figure legend.
Both figures are excellent.
Author Response
Responses to REVIEWER 2:
The review paper "The Impact of the Nervous System on Arteries and the Heart: The Neuroimmune Cardiovascular Circuit Hypothesis", written by authors Mohanta SK, Sun T, Wang Z, Zhang X, Yin Z, Weber C, Habenicht AJR is up-to-date, comprehensive and scientifically valuable contribution for integrative approach to understanding of cardiovascular physiology and diseases. Concept of "neurocardiovascular (diseases, circuits...)" is successfully substituted by "neuroimmune cardiovascular (circuit hypothesis)".
R2: Principles like brain morpho-functional organization were discussed. "...the brain can be viewed as a storage hub of trillions of neurons tasked to control peripheral tissue homeostasis." (page 2/12, lines 4 &5) . It is my opinion that also the concept of "homeodynamics" and/or adaptability (Goldberger AD, 2006, doi 10.1513/pats200603-028MS; Matić Z et al, 2020, doi 10.3389/fphys.2020.00024) should be mentioned and discussed, especially in association with learning, memory and transfer of information. In the next sentence (page 2/12, line 5-9), integration centers were described as neural networks supporting the survival of multicellular organisms "during multiple acute and chronic stress". It is my opinion that the term "stress" should be substituted by the term "environmental and internal body challenges". Modern definitions of stress include coping capacity of the organism, i.e. stress as the state of the organism exposed to environmental demands BEYOND coping capacity of the organism.
RESPONSE: We integrated Reviewer # 2 suggestions in the revised text.
R2: Concept of bidirectional (i.e. reciprocal) tripartite neuroimmune cardiovascular circuit is is very well presented as PRINCIPLE. Still, an important issue of PATTERN, individually specific for healthy individuals (genetically conditioned), behaviorally specific (Porta A et al, 2012, doi 10.1016/j.compbiomed.2011.04.019; Matić Z et al, 2020, doi 10.3389/fphys.2020.00024) or disease specific (Radovanović N et al., 2018, doi 10.3389/fphys.2018.00165) is missed to be mentioned. The pattern (i.e. intensity of the each information transfer within bidirectional communication) deserves to me mentioned.
RESPONSE: Thank you for discussing these important aspects. We now introduced your points in the revised manuscript (Page 4, last paragraph).
R2: Please make a new paragraph between figure legend and the following body text. After both figure legends text proceeds in continuation with the text of figure legend.
RESPONSE: Thank you. We now inserted the spaces.
R2: Both figures are excellent.
RESPONSE: Thank you.
Reviewer 3 Report
-It would be helpful to briefly mention the significance of the ABC and HBC individually before introducing the idea of them forming a larger CBC. This would provide context for readers who may not be familiar with these sub-circuits.
-It would be valuable to discuss the potential limitations or challenges associated with studying and targeting the CBC. This would provide a more comprehensive perspective and acknowledge any potential obstacles that researchers in the field may face.
-The abstract could benefit from a clearer structure, with distinct paragraphs dedicated to introducing the ABC and HBC, discussing the formation of the CBC, highlighting the potential therapies, and presenting the Neuroimmune Cardiovascular Circuit Hypothesis.
-Select keywords according to MeSH.
-Line 28 has 2 "that".
-Provide empirical evidence, offer practical implications, address potential limitations, and encourage further research directions in the introduction.
-Provide supporting evidence, address potential mechanisms, address limitations and challenges, explore therapeutic implications, discuss clinical relevance, propose future research direction in more details for part 5.
Minor editing of English language required
Author Response
Responses to REVIEWER 3:
R3: The abstract could benefit from a clearer structure, with distinct paragraphs dedicated to introducing the ABC and HBC, discussing the formation of the CBC, highlighting the potential therapies, and presenting the Neuroimmune Cardiovascular Circuit Hypothesis.
RESPONSE: We now extended the abstract based on the content of the manuscript, and made attempts to provide a clearer structure.
R3: Select keywords according to MeSH.
RESPONSE. We have corrected the keywords according to MeSH.
R3: -Line 28 has 2 "that".
RESPONSE. Thank you. We have corrected the error.
R3:-Provide empirical evidence, offer practical implications, address potential limitations, and encourage further research directions in the introduction.
RESPONSE: We made attempts to improve the manuscript regarding these points. However, we are limited in the scope of this review because of space constraints but refer to a comprehensive review published in Circulation Research (Ref# 1; Mohanta et al. 2023). The point of R3 to indicate further research directions: We prefer to meet the request of R #1 to place further research directions at the end of this review rather than in the introduction.
R3:-It would be helpful to briefly mention the significance of the ABC and HBC individually before introducing the idea of them forming a larger CBC. This would provide context for readers who may not be familiar with these sub-circuits.
RESPONSE: We followed this concept in the revised manuscript.
R3: -It would be valuable to discuss the potential limitations or challenges associated with studying and targeting the CBC. This would provide a more comprehensive perspective and acknowledge any potential obstacles that researchers in the field may face.
RESPONSE: We have made attempts to meet this request of R #3 in the revised manuscript.
R3:-Provide supporting evidence, address potential mechanisms, address limitations and challenges, explore therapeutic implications, discuss clinical relevance, propose future research direction in more details for part 5.
RESPONSE: We have made attempts to meet the requests of R #3 in both section 5 and 6 of the revised manuscript.
Round 2
Reviewer 1 Report
All my suggestions were taken into account. In my opinion, the article can be accepted in its current form
Reviewer 3 Report
The paper improved very well and it is acceptable.